# Management of Patients with Diabetic Macular Edema Switched from Dexamethasone Intravitreal Implant to Fluocinolone Acetonide Intravitreal Implant

**DOI:** 10.3390/pharmaceutics14112391

**Published:** 2022-11-05

**Authors:** Stéphanie Baillif, Pascal Staccini, Michel Weber, Marie-Noëlle Delyfer, Yannick Le Mer, Vincent Gualino, Laurence Collot, Pierre-Yves Merite, Catherine Creuzot-Garcher, Laurent Kodjikian, Pascale Massin

**Affiliations:** 1Department of Ophthalmology, Hôpital Pasteur 2, Centre Hospitalier Universitaire de Nice, Université Côte d’Azur, 30 Voie Romaine, 06000 Nice, France; 2INSERM DR2 U1065, C3M, 151 Avenue Saint-Antoine de Ginestière, 06024 Nice, France; 3Research Unit RETINES (Risks, Epidemiology, Territories, Information and Education for Health), Université Côte d’Azur, Faculté de Médecine, 28 Avenue de Valombrose, CEDEX 2, 06107 Nice, France; 4Department of Ophthalmology, Centre Hospitalier Universitaire de Nantes, 44000 Nantes, France; 5Department of Ophthalmology, Bordeaux University Hospital, 33000 Bordeaux, France; 6INSERM, BPH, UMR1219, Bordeaux University, 33000 Bordeaux, France; 7Department of Ophthalmology, A. de Rothschild Ophthalmologic Foundation, 75019 Paris, France; 8Clinique Honoré Cave, Department of Ophthalmology, 82000 Montauban, France; 9Ophthalmology Department, AP-HP, Hôpital Lariboisière, Université de Paris, 75014 Paris, France; 10Centre Médico-Chirurgical de Chaumont, 17 Avenue des États-Unis, 52000 Chaumont, France; 11Centre D’ophtalmologie, 44 Avenue de Lattre de Tassigny, 13090 Aix-en-Provence, France; 12Department of Ophthalmology, Dijon University Hospital, 14 Rue Gaffarel, 21000 Dijon, France; 13Department of Ophthalmology, Croix-Rousse University Hospital, Hospices Civils de Lyon, 69002 Lyon, France; 14UMR-CNRS 5510 Matéis, University of Lyon, 69622 Villeurbanne, France; 15Ophthalmic Centre of Breteuil, 14 avenue de Breteuil, 75007 Paris, France

**Keywords:** diabetic macular edema, dexamethasone implant, fluocinolone acetonide implant, intravitreal therapy

## Abstract

To assess anatomical and functional outcomes after switching from dexamethasone implant (DEXi) to fluocinolone acetonide implant (FAci) in 113 diabetic macular edema eyes, a multicentric retrospective observational study was conducted. Seventy-five eyes (73.5%) were switched 1–8 weeks after their last DEXi. The mean best-corrected visual acuity improved to 59.8 letters at month 4 and remained stable during the follow-up. The mean central macular thickness (CMT) significantly decreased during the follow-up, with a minimum of 320.9 μm at month 3. The baseline CMT was higher in eyes that received the last DEXi >8 weeks versus <8 weeks before the first FAci (*p* < 0.021). After FAci injection, additional treatments were needed in 37 (32.7%) eyes. A longer diabetes duration (*p* = 0.009), a longer time between the last DEXi and the first FAci (*p* = 0.035), and a high baseline CMT (*p* = 0.003) were risk factors for additional treatments. The mean intraocular pressure was <19 mmHg at all timepoints, with no difference between eyes receiving the last DEXi ≤8 weeks or >8 weeks before the switch. Switching from DEXi to FAci in DME is effective and safe. A short time between the last DEXi and the first FAci reduced CMT fluctuations and the need for early additional treatments.

## 1. Introduction

Diabetic Macular Edema (DME) is a leading cause of vision loss in patients with Diabetes Mellitus (DM) [1,2]. It affects 6.8% of patients with DM [3], and up to 24% of patients will develop DME within 10 years of diagnosis [4]. If left untreated, DME may lead to a loss of visual acuity (VA) ≥2 lines in 2 years [5].

DME is a multifactorial condition. Sustained hyperglycemia upregulates several vasoactive and proinflammatory factors, including VEGF-A [6]. The role of inflammation is also well established with the involvement of leukostasis, the upregulation of inflammatory cytokines, macrophage infiltration, leukocyte adhesion, and complement activation [7].

The current first-line pharmacological standard of care for clinically significant DME targets VEGF-A [8]. However, in randomized clinical trials, 31.6–65.6% of patients with DME respond suboptimally to anti-VEGF therapy [9,10]. In real-life studies, the visual outcomes may be poorer than those achieved in clinical trials, due to undertreatment as an additional factor [11,12]. Indeed, anti-VEGF therapy requires multiple injections, resulting in higher treatment burden and lower treatment adherence in diabetic patients [13].

Intravitreal corticosteroids are often prescribed as second-line treatments for DME, due to their known steroid-induced sides effects, such as intraocular pressure (IOP) elevation and cataract formation. Their use in DME is justified by the fact that corticosteroids have a wide range of anti-inflammatory properties: they reduce the expression of VEGF-A [14], suppress the influx of leukocytes in the retina, inhibit the expression of intercellular adhesion molecule-1, decrease paracellular permeability, and increase tight junction integrity and transepithelial resistance [15].

Two intravitreal corticosteroids are approved for DME treatment. The dexamethasone (DEX) implant (OZURDEX^®^, Allergan Ltd., Irvine, CA, USA) is a biodegradable copolymer of polylactic-co-glycolic acid that releases 0.7 mg of dexamethasone over a 6-month period [16], with a high drug release rate in the first 2 months, followed by a lower release rate for 4 months [17]. However, the mean retreatment time is shorter than 6 months [18,19,20]. Indeed, an anatomical recurrence of DME, rapidly followed by a functional impairment, may occur about 4–5 months after DEX injection [21].

The fluocinolone acetonide (FAc) implant (ILUVIEN^®^, Alimera sciences Inc., Alpharetta, GA, USA) is a non-biodegradable intravitreal implant that releases 0.2 μg of fluocinolone acetonide/day over a 3-year period. It is a slow-release implant, showing a zero-order release kinetics with a low peak concentration and a steady-state concentration for 36 months [22]. 

The efficacy and safety of both DEX and FAc implants have been demonstrated in randomized clinical trials with a substantial visual benefit and a reduction in central macular thickness (CMT) [16,23,24,25]. Real-world studies have reported similar results with good anatomical and functional outcomes [21,26,27,28,29].

Switching from an anti-DME therapy to another is based on many factors, such as a suboptimal response, the presence of absolute or relative contraindications, the occurrence of adverse events, patients/physicians’ preferences [30], patient’s compliance, or country-specific recommendations [31,32].

Intravitreal injections of FAc may be prescribed directly after anti-VEGF injections [19,27,28] or after additional intravitreal steroid injections (dexamethasone implant or off-label triamcinolone injection) [33,34,35,36].

The reasons for switching from DEX to FAc implants may be a suboptimal response to intravitreal DEX, a short-term effect of DEX injections, or a steroid challenge before FAc administration. Indeed, the FDA has approved the FAc implant in DME patients previously treated with a course of corticosteroids in the absence of any clinically significant rise in IOP. A few small studies have investigated the switch from DEX to FAc implants [33,34,35,36]. However, the management of this switch, particularly in regard to the timing of the FAc injection or the need for additional treatment, has not been fully detailed.

The aims of this multicentric retrospective observational study were to determine the anatomical and functional responses in a large series of DME patients switched from short-acting DEX implant to longer-acting FAc implant. The management of the switch was described and its consequences in terms of safety and efficacy were assessed.

## 2. Methods

### 2.1. Study Design

This was a real-world multicentric retrospective observational study conducted in patients with DME who received at least one intravitreal DEX implant injection prior to FAc implant injection. In all participating sites, patients’ charts were reviewed, and data were collected. The participating sites were all members of the CFSR (Club Francophone des Spécialistes de la Rétine), which is a French society of retina specialists. Each member received a structured template for data collection. All DME patients who received at least one FAc implant injection and with a follow-up of at least 4 months were eligible. Information was collected between April 2020 and March 2021. Patients with other causes of macular edema (retinal vein occlusion, uveitis, post-surgical macular edema, tractional macular edema, inherited retinal dystrophies, and drug-induced macular edema) were excluded from the study. Patients with vitreomacular traction or macular hole were also excluded.

The study was conducted in accordance with the principles outlined in the declaration of Helsinki and was approved by the ethics committee of the French Ophthalmology Society (IRB00008855). 

Demographics (age and gender), medical history (duration and type of DM, insulin therapy, presence of associated high blood pressure, and glycated hemoglobin (HbA1c) level), and ophthalmologic history (stage of diabetic retinopathy, DME duration, previous treatments for DME, history of cataract surgery or vitrectomy with or without epiretinal membrane peeling, history of open-angle glaucoma, and use of IOP-lowering medications) were collected at baseline.

The best-corrected visual acuity (BCVA; ETDRS letter score), CMT, and IOP were recorded at baseline (prior to FAc intravitreal injection) and at each follow-up visit, performed at the participant’s site discretion. The number of follow-up visits was recorded.

After FAc injection, ocular adverse events (cataract surgery, endophthalmitis, raised IOP, and changes in IOP-lowering procedures) and additional ocular treatments received after FAc injection were recorded.

### 2.2. Statistical Analysis

The statistical analysis was performed by using Minitab^®^ (20.3, © 2021 Minitab, LLC, Pennsylvania, PA, USA) and XLSTAT (Addinsoft, 2022). All quantitative variables are presented as a mean ± standard deviation. Additional distribution parameters are presented, including the median and range. Due to non-normal distributions, the Spearman rank-order correlation coefficient was used to test the association between continuous variables. Comparisons between quantitative (age, DM duration, DME duration, HbA1c level, BCVA, IOP, CMT, and number of prior DEX injections) and categorical variables (time between the last DEX injection and the first FAc injection ≤8 weeks and need the for additional treatment or not) were performed by using the non-parametric Kruskal–Wallis test. The Chi-squared test was used to compare categorical variables. For the multivariate analysis, a general linear model and a binary logistic regression were used depending on the nature of the explained variable. A *p* < 0.05 was considered statistically significant. 

## 3. Results

### 3.1. Study Population

A total of 113 eyes from 30 participating sites were included. Twenty-seven patients received bilateral FAc injections. The mean follow-up duration was 7.6 months (range: 4–12 months).

The baseline demographics and ocular characteristics are summarized in Table 1. Patients’ mean age was 69.8 ± 10.2 years; 51.3% were women. Regarding DM type and duration, 89.3% of patients had type 2 DM for a mean duration of 21.9 ± 12.7 years. High blood pressure was found in 70.3% of patients.

The mean DME duration was 71.8 ± 48.2 months (range: 15–360 months), and 5.6% of patients had proliferative diabetic retinopathy. Moreover, 64.8% had a history of panretinal photocoagulation. Most patients were already pseudophakic (91.2%). Thirty eyes had undergone vitrectomy, and the epiretinal membrane had been peeled in nineteen cases.

### 3.2. Prior DME Treatment

All the eyes (n = 113, 100%) received intravitreal DEX implant injections prior to FAc injection. The mean number of DEX injections was 6.3 ± 4.5 (range: 1–21), with a mean time between two injections of 14.8 ± 4.5 weeks (Table 2). The mean time between the last DEX injection and the first FAc injection was 11.12 ± 22 weeks (range: 1–163). The time between the last DEX injection and the first FAc injection ranged between 1 and 4 weeks in 41 eyes (40.2%), ranged between 5 and 8 weeks in 34 eyes (33.3%), and was longer than 8 weeks in 27 eyes (26.5%). Thus, 75 eyes (73.5%) were injected with FAc ≤8 weeks after their last DEX injection.

Prior to DEX and FAc injections, 27 out of 103 eyes (26.2%) had been treated with macular laser therapy, and 97 out of the 113 eyes (85.8%) had received prior anti-VEGF treatments (75.3% with ranibizumab (n = 72/97), 50.5% with aflibercept (n = 49/97) and 10.3% with bevacizumab (n = 10/97)) (Table 2).

Intravitreal triamcinolone was prescribed in 13 out of the 113 eyes (11.5%), while 5 out of the 113 eyes (4.4%) received periocular triamcinolone.

### 3.3. Visual Acuity

Prior to the first FAc injection, the mean baseline BCVA was 54.1 ± 17.8 ETDRS letters (n = 113), and it improved up to 59.8 letters at month 4 (n = 51). The mean baseline VA was 54.8 ETDRS letters and 54.5 ETDRS letters, respectively, in eyes that received the last DEX injection ≤8 weeks and >8 weeks before the first FAc injection. The mean baseline BCVA was <70 ETDRS letters in 86 out of the 113 eyes (76.1%). The mean BCVA remained stable throughout the follow-up (Figure 1 and Table 3). The mean gain in VA was the highest at month 4 (+5.3 ETDRS letters) and the lowest at month 12 (−2.8 ETDRS letters) (Figure 1). A gain >15 ETDRS letters was observed in 9%, 15%, 20%, and 20% of eyes at month 1, month 3, month 6, and month 12, respectively (Table 3). In the univariate analysis, the age (*p* = 0.029), the presence of high blood pressure (*p* = 0.042), and the DME duration (*p* = 0.008) correlated with the change in VA. In the multivariate analysis, only the DME duration correlated with the change in VA (*p* = 0.026).

### 3.4. Retinal Thickness

Prior to the first FAc injection, the mean CMT was 454.7 ± 196.7 μm (median: 410.5 μm). The CMT significantly decreased during the follow-up to reach 320.9 μm at month 3 (Figure 2). The mean decrease in CMT was the largest at month 3 and the smallest at month 2 (Figure 2). At baseline, 22 out of 108 eyes (20.4%) had a CMT ≤300 μm, and this proportion increased to 38 out of 73 eyes (52%), 25 out of 45 eyes (55%), and 18 out of 45 eyes (40%) at month 1, month 3, and month 6, respectively. The baseline CMT was higher in eyes that received the last DEX injection >8 weeks compared to ≤8 weeks before the first FAc injection (*p* < 0.021). However, during the follow-up, the mean CMT did not differ between patients who received the last DEX injection ≤8 weeks or >8 weeks prior to the first FAc injection (Figure 3).

### 3.5. Additional Treatments

After FAc injections, 37 out of the 113 eyes (32.7%) needed additional treatments for DME, including macular laser therapy in 3 out of 37 eyes (8.1%), intravitreal injection of anti-VEGF in 18 out of 37 eyes (48.6%), and dexamethasone intravitreal implant in 15 out of 37 eyes (40.5%). One patient was treated with subconjunctival triamcinolone. 

The mean time to additional treatment was 113.27 ± 94.70 days (range: 0–462), with a first quartile, median, and third quartile time to additional treatment of 45.5, 102.0, and 151.0 days, respectively. Two patients received an intravitreal injection of anti-VEGF on the same day as their FAc injection. 

During the follow-up, 25 out of 37 eyes (67.5%) received two additional treatments, 21 out of 37 eyes (56.7%) received three additional treatments, and 9 out of 37 eyes (24.3%) received four additional treatments. 

During the follow-up, there was no difference in BCVA between patients who received or not additional treatments (Figure 4). The mean CMT was lower in patients who did not receive any additional treatment compared to patients for whom additional treatments were needed (Figure 4). There was no difference in CMT between patients who received an anti-VEGF or dexamethasone implant as an additional treatment.

The variables associated with the need for additional treatments are summarized in Table 4. A longer DM duration (*p* = 0.009), a longer time between the last DEX injection and the first FAc injection (*p* = 0.035), and a high baseline CMT (*p* = 0.003) were considered risk factors for additional treatments after FAc intravitreal injection.

The variables associated with the mean time to additional treatment after FAc injection were the number of prior DEX implants injected (*p* < 0.05), the presence of induced high blood pressure (*p* = 0.044), and the grade of diabetic retinopathy (*p* < 0.05). 

### 3.6. Safety

At baseline, 23 out of 90 eyes (25.6%) were treated with IOP-lowering medications (Table 1). Among them, 52.2%, 39.1%, and 8.7% were treated with IOP-lowering drugs as monotherapy, dual therapy, and triple therapy, respectively. Four out of eighty-five eyes had undergone selective laser trabeculoplasty prior to FAc injection. No eyes had undergone incisional glaucoma surgery prior to FAc injection. 

The mean baseline IOP was 19.0 ± 4.5 mmHg (range: 9–26; median: 19.0 mmHg). The mean IOP was <19 mmHg throughout the follow-up (Figure 5). During the follow-up of the 113 eyes, the IOP was >21 mmHg in 18 eyes (15.9%), >25 mmHg in 12 eyes (10.6%), and >30 mmHg in 4 eyes (3.5%). In the 18 eyes with an IOP >21 mmHg, 11 (61.1%) were injected with FAc ≤8 weeks after their last DEX injection. In these 18 eyes, 9 (50%) were already treated with IOP-lowering drops before the FAc injection. Patients with an IOP ≥30 mmHg received their last DEX injection 12, 18, 40, and 45 weeks, respectively, before their FAc injection. Three of these eyes were treated for ocular hypertension before the FAc injection (two with IOP-lowering agents as dual therapy and one with IOP-lowering agents as triple therapy).

During the follow-up, 29 out of 90 eyes (32.2%) received anti-glaucoma medications. Among them, six were naïve of IOP-lowering drops, and the mean time to initiate IOP-lowering medication after the FAc injection was 4.8 months. For the other 23 patients, an additional drug was added to their initial IOP-lowering treatment. 

During the follow-up, one eye was treated with selective laser trabeculoplasty and one with microinvasive glaucoma surgery: these two eyes were already treated with IOP-lowering drops (one as triple therapy and one as dual therapy) before the FAc injection. No filtering glaucoma surgery was performed during the follow-up.

At month 1, the IOP values correlated with the baseline IOP values (prior to the FAc injection) (*p* = 0.047) and with the use of IOP-lowering drops at baseline (*p* = 0.031). At month 3, the IOP values correlated with the use of IOP-lowering drops at baseline (*p* = 0.034). From month 1 to month 4 and at month 8, the IOP values did not correlate with the number of DEX implants received prior to the FAc injection or with the time between the last DEX implant injection and the first FAc injection. During the follow-up, the evolution of the IOP was similar between patients who received the last DEX implant injection ≤8 weeks and >8 weeks prior to the FAc injection (Figure 6).

No patient developed an endophthalmitis during the follow-up. Prior to the FAc injection, 10 out of the 113 eyes (8.8%) were phakic. Among these eyes, one eye underwent cataract surgery 8 months after the FAc injection.

## 4. Discussion

The main objectives of this retrospective study were to assess the efficacy and safety of FAc intravitreal implants in patients with chronic DME previously treated with DEX intravitreal implants and to describe how the switch between both corticosteroid implants was managed.

In most of the studies assessing FAc injections in DME, the patients were previously treated with anti-VEGF therapy and less frequently with intravitreal steroids (off-label triamcinolone or dexamethasone implant) [24,26,27,28,31,37,38,39]. In any of these studies, the last treatment given just prior to the FAc injection was clearly specified. Only McCluskey et al. have mentioned that, among prior anti-DME treatments, the most recent intervention was an anti-VEGF agent in 38.9% of eyes, and dexamethasone in 44.4% of eyes [19], but no subgroup analysis of the VA or CMT according to the last treatment received prior to the FAc injection was performed. The authors, however, found a statistically significant improvement in macular volume between baseline and the last observation in eyes for which the most recent intervention was steroid therapy (*p* = 0.002), and specifically dexamethasone (*p* = 0.006) [19]. Thus, very limited data on the functional outcomes depending on the last treatment received (i.e., intravitreal anti-VEGF or steroids) are available.

More recently, four studies including 22 [35], 25 [33], and 44 eyes [34,36], respectively, have investigated whether prior treatment for DME with intravitreal corticosteroids (dexamethasone implant) could affect the outcome of the FAc implant. Rehak et al. compared eyes with DME achieving a suboptimal response to anti-VEGF therapy that were switched to the FAc implant either directly (n = 24 eyes) or indirectly (first switch to dexamethasone, n = 25 eyes) [33]. In the other three studies, there were no comparative groups, and all eyes were switched to FAc just after the DEX intravitreal implant. In our retrospective non-comparative study, 113 eyes were included, and this is the largest series of eyes switched from DEX to FAc reported to date. We observed that treatment with the FAc implant was associated with BCVA stability during the follow-up. A mean peak VA gain of +5.3 letters was, however, reached 4 months after the FAc injection. These BCVA results are in line with the findings previously reported in clinical trials and case series, showing VA improvement or maintenance after FAc injection in chronic DME eyes, regardless of the last treatment received prior to the FAc injection. Indeed, most real-world studies have reported a BCVA improvement ≥5 letters after the FAc injection, with a mean peak VA gain of +8.7 letters after 11.3 months [29]. Rehak et al. found similar improvements in BCVA in their two groups of patients regardless of whether they were switched directly or indirectly (after DEX implant) from an anti-VEGF to the FAc implant [33]. However, when comparing our results to those obtained in series switched directly from DEX to FAc, we observed that our BCVA gains were much lower. Vaz-Pereira et al. obtained a mean VA gain from baseline of +6.82 letters at month 1 and +13.02 letters at month 6 [36], while Elbarky et al. obtained a mean VA gain from baseline of +22.8 letters at month 3 and +25.5 letters at month 12 [35]. These discrepancies could be due to differences in patients’ baseline characteristics, such as a shorter DME duration [36], a lower baseline VA [35,36], and variable clinical responses to prior DEX injections [34]. However, these improvements could also be related to the conditions of the switch from the DEX implant to the FAc implant.

Indeed, in our study, 73.5% of eyes received their FAc injection ≤8 weeks after their last DEX injection, meaning that the intravitreal concentration of dexamethasone was maximum when the FAc injection was given [40]. As the pattern of mean change in VA and CMT observed reflects the kinetics of dexamethasone release [16,25], improvements in BCVA and CMT were therefore almost maximum before the FAc injection [41]. Thus, the benefit of FAc in terms of BCVA was expected to be lower in these eyes than in eyes experiencing DME recurrence before FAc injection [33,36]. Moreover, in our study, we observed an increase in VA during the follow-up, showing that FAc could offer additional benefits in patients treated with DEX implants.

This benefit in terms of VA could be due to a reduced variability of the retinal thickness at the time of the switch and during the follow-up after FAc injection. Indeed, reduced variations in retinal thickness due to the recurrence of DME have been shown to lead to long-term visual improvement and stability [42]. Clinically, both randomized controlled studies [22,24] and real-world studies have confirmed that the FAc implant allows us to stabilize the CMT [34,43,44,45,46,47], with substantially reduced edema recurrence and variations over a period of 3 years.

In our study, the mean CMT significantly decreased from month 1 to month 12, reaching its minimum 3 months after the FAc injection. The mean baseline CMT was significantly lower in eyes that received the last DEX injection ≤8 weeks compared to >8 weeks before the FAc injection. However, during the follow-up, no differences were observed between the two groups. As mentioned, the difference in the mean baseline CMT could be due, among other factors, to the conditions of the switch with a short mean time between the last DEX injection and the FAc injection. In eyes with a time between the last DEX injection and the FAc injection ≤8 weeks, a mean number of 6.32 prior DEX implants were injected. Thus, physicians were able to observe the kinetics of the BCVA and CMT after each DEX injection and to schedule the switch in order to avoid any DME recurrence between both treatments. Indeed, even if FAc and DEX implants are both fluorinated glucocorticoids, they harbor molecular differences leading to differences in potency, solubility, distribution, and pharmacokinetic profile in the eye [40,48,49]. The DEX implant has a short pulse kinetics, with a significant burst of drug release for about 2 months before an exponential decline in release over 6 months [40]. The FAc implant has a slower release that is sustained over a longer period of time [17]. In rabbits, FAc levels have been shown to nearly reach the steady state 3 months after the injection and then gradually decrease over 24 months in most ocular tissues [49]. In humans, steady-state aqueous levels are reached within about 6 months after the injection [22]. At month 12, aqueous concentrations of FAc slightly decrease and drop between 12 and 36 months [22]. Clinically, a reduction in CMT is observed from 7 days post-FAc injection [24,47], with a substantial reduction from month 1 to month 3, and it then remains stable for 24 months [23,37]. Kodjikian et al., in their systematic review of real-world studies, have observed a maximum decrease in CMT from baseline by −34.3% at month 16.6 [29]. Thus, as the peak in efficacy after FAc injection is delayed compared to that of DEX, the switch from DEX to FAc could be anticipated and adjusted according to the history and response to previous treatments of the eye. The objective would be to limit CMT variations by maintaining effective steroid intravitreal levels when switching from DEX to FAc [41].

In our study, 32.7% of eyes received additional treatments after the FAc injection. This proportion is in line with the published literature, where 55.6–67.5% of eyes did not require additional treatment after FAc injection [19,28,29,38,50]. The time between the FAc injection and the administration of first additional treatment is highly variable with a large standard deviation [27,34]. Kodjikian et al. found that almost 30.0% of patients needed one additional intravitreal treatment during the follow-up, with a mean time between the FAc injection and the administration of additional treatment of 15.4 months [29]. In our cohort, the mean time to additional treatment was 113.27 ± 94.70 days, with first and median quartile times to additional treatment of 45.5 and 102.0 days, respectively. These results showed the presence of two populations of eyes: those requiring early additional treatment during the first 3 months after FAc injection when the clinical efficacy of FAc is not yet maximum, and those requiring late additional treatment when the steady state of FAc is reached. Moreover, the time between the last DEX injection and the first FAc injection was associated with the risk of needing additional treatment (*p* = 0.035). A high baseline CMT (*p* = 0.003), which was significantly higher in eyes that received the FAc injection >8 weeks after the last DEX injection, was another risk factor for the need for additional treatment after FAc injection. Thus, we could assume that eyes with a shorter time between the last DEX injection and the FAc injection could need less additional therapy during the first three months after FAc injection, when the intravitreal dexamethasone activity counterbalances an increasing but still not completely effective intravitreal activity of FAc.

During the follow-up, 67.5%, 56.7%, and 24.3% of the 37 eyes treated with one additional therapy after the FAC injection required respectively two, three, and four additional intravitreal injections. These patients might not be good anatomical responders to Fac, since rescue therapy was needed throughout the follow-up. However, we did not know if these patients were considered good morphological responders to intravitreal dexamethasone: the kinetics of CMT reduction after DEX injections and the mean time between each prior DEX injection were not recorded. Such data would have been of interest knowing that the response to prior DEX injections could help to anticipate the morphological response to subsequent FAc injection and the need for additional anti-DME therapy. Indeed, Cicinelli et al. observed that only eyes with a good morphological response to DEX (defined as a CMT decrease by at least 95 μm 1 month after the first DEX implant injection) showed a significantly reduced CMT after FAc injection (*p* < 0.001) [34]. Similarly, a higher proportion of poor morphological responders to DEX received additional treatments after FAc injection compared to good morphological responders (*p* = 0.006).

Safety issues were of major concern in this study, since 73.5% of eyes received two different intravitreal corticosteroids over a 2-month period. The timing of the IOP rise with intravitreal steroids suggests that elevated IOP reflects the drug release into the eye and returns to baseline as the drug concentration decreases [51]. In DEX implant randomized controlled trials, the highest change in IOP after DEX implant injection occurred within 3 months after implantation, and it subsequently dropped by month 6 [16]. After FAc injection, the FAME study showed that the mean IOP increased between baseline and the months 3–12 [37]. Real-life studies have shown comparable results with an increase in mean IOP during the first 12 months, followed by a decrease from month 12 to month 30 [38,39]. Augustin et al. found the largest mean changes in IOP during the first 6 months after FAc implantation, with a peak change by +2.5 mmHg at month 6 [38]. In our study, the safety profile related to the IOP was good. The mean IOP remained <19 mmHg during the follow-up and the proportions of patients with a recorded IOP >21, >25 or >30 mmHg at any visit were 15.9%, 10.6%, and 3.5%, respectively, and these proportions were less than those observed in the FAME study or in other real-life studies [26,27]. During the follow-up, the mean IOP was not different between patients who received their last DEX implant ≤8 weeks and >8 weeks before the first FAc injection.

In our study, 32.2% of eyes received anti-glaucoma medications after FAc injection. Among these eyes, IOP-lowering treatment was initiated in only 6 eyes during the follow-up, while the other 23 eyes were already treated with IOP-lowering drugs at baseline. Three out of the four eyes with an IOP ≥30 mmHg and the two eyes requiring selective laser trabeculoplasty or microinvasive glaucoma surgery were already treated for ocular hypertension at baseline. These results highlight the need for carefully selecting patients based on the occurrence of a prior steroid-induced IOP-elevation event before FAc injection. Several studies have shown that the absence of clinically significant IOP elevation after prior intravitreal steroid use has a very strong positive predictive value for the absence of IOP elevation post-FAc implant injection [26,28,34,39,43,47,52]. Therefore, a steroid challenge is recommended to assess the risk of a steroid-induced IOP rise in response to FAc injection.

Our study has several limitations. The main one relates to the collection and reporting of retrospective real-life data, which were not consistently available for all patients at all timepoints. This unavailability of data was increased by the COVID-19 pandemic since patients were not followed as regularly as they should have been. Visits were missed or delayed, explaining the distribution of data during the follow-up. Moreover, our follow-up was short, but it was, however, sufficient to explore the transitional period of the switch from the DEX implant to the FAc implant.

Another limitation of our study was the use of the CMT alone to monitor the anatomical response to FAc injection. Indeed, the CMT measured at specific timepoints could not accurately reflect the post-injection variability in retinal thickness. Moreover, a poor correlation between the CMT and the visual outcomes was previously reported [53]. Changes in retinal thickness amplitude or the standard deviation of the central subfield thickness could be more accurate to characterize retinal thickness stability [43]. They have been shown to better correlate with VA [28].

To conclude, this study confirmed the efficacy of the FAc implant in DME patients previously treated with DEX implants. In most patients, the timing of the switch from the last DEX implant to the FAc implant was scheduled in order to avoid DME recurrence. A short time between the last DEX injection and the first FAc injection reduced CMT variations during the transitional phase and could reduce the need for early additional treatments. The safety profile was reassuring even in patients injected with DEX and FAc implants over a 2-month period. Further studies are needed to clarify the best timing for switching from DEX to FAc implants in terms of efficacy and safety.

## Figures and Tables

**Figure 1 pharmaceutics-14-02391-f001:**
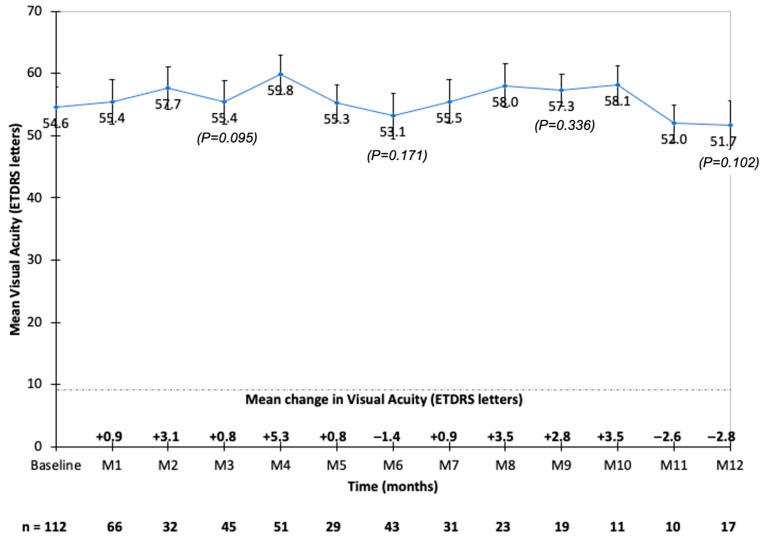
Mean best-corrected visual acuity (BCVA) and mean change in BCVA during the follow-up after fluocinolone acetonide implant injection.

**Figure 2 pharmaceutics-14-02391-f002:**
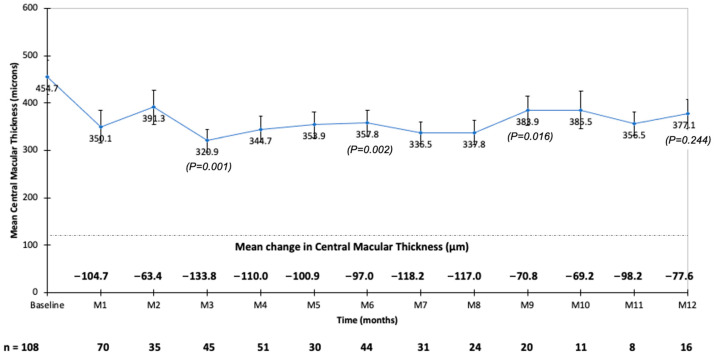
Mean central macular thickness (CMT) and mean change in CMT during the follow-up after fluocinolone acetonide implant injection.

**Figure 3 pharmaceutics-14-02391-f003:**
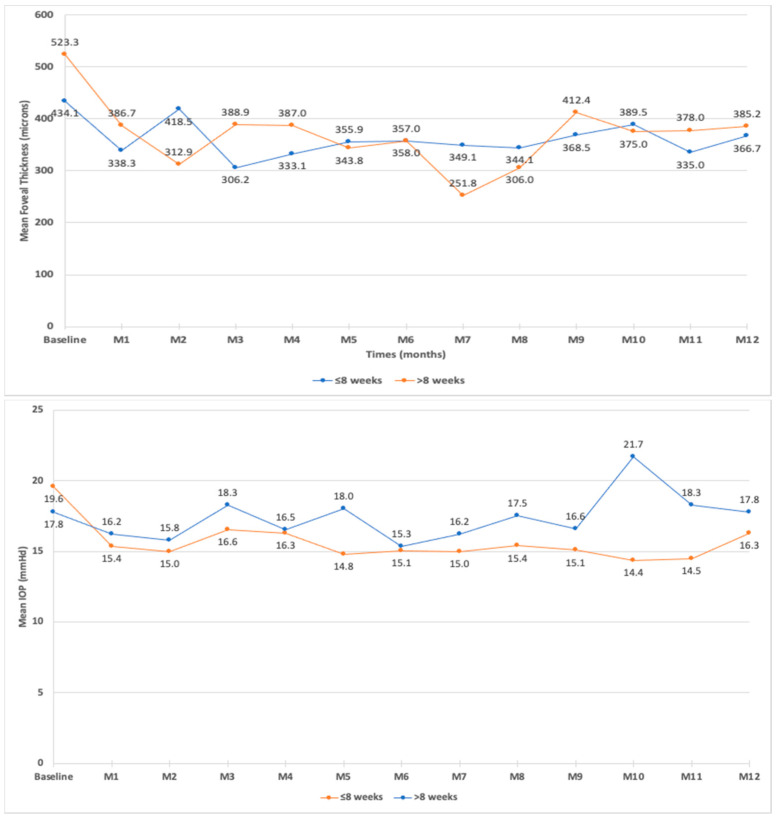
Mean central macular thickness (CMT) and mean intraocular pressure (IOP) in patients who received the last dexamethasone intravitreal implant injection less than 8 weeks or more than 8 weeks prior to fluocinolone acetonide intravitreal implant injection.

**Figure 4 pharmaceutics-14-02391-f004:**
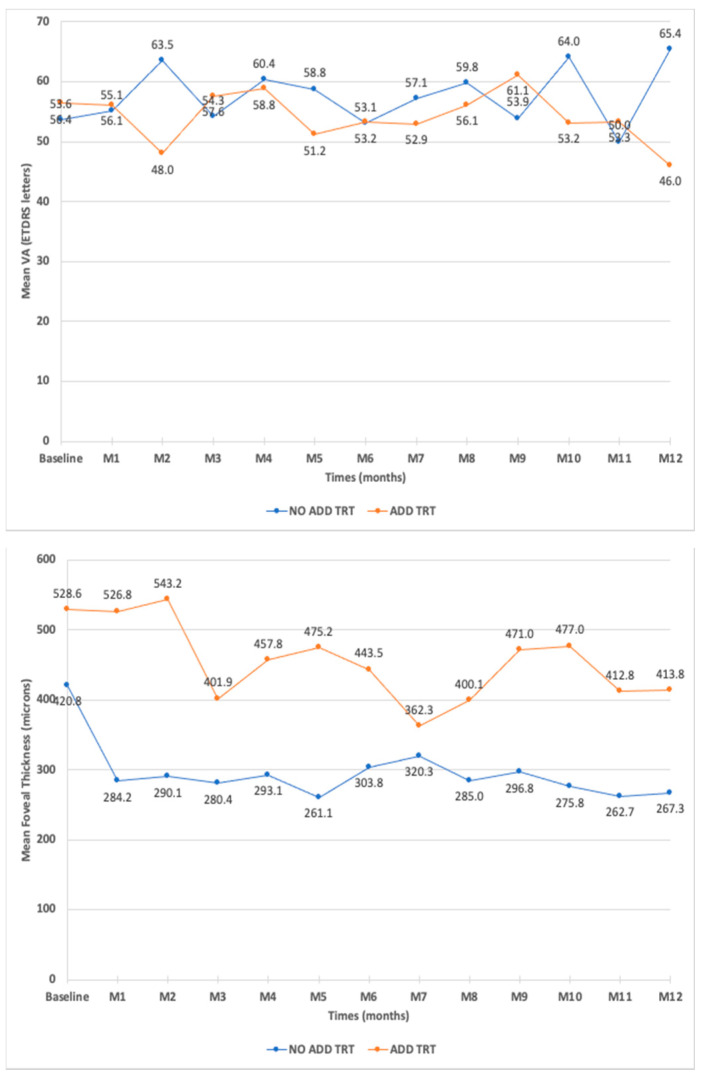
Mean best-corrected visual acuity (BCVA) and mean central retinal thickness (CMT) in eyes that needed additional treatments compared to eyes that did not need additional treatments after fluocinolone acetonide intravitreal injection.

**Figure 5 pharmaceutics-14-02391-f005:**
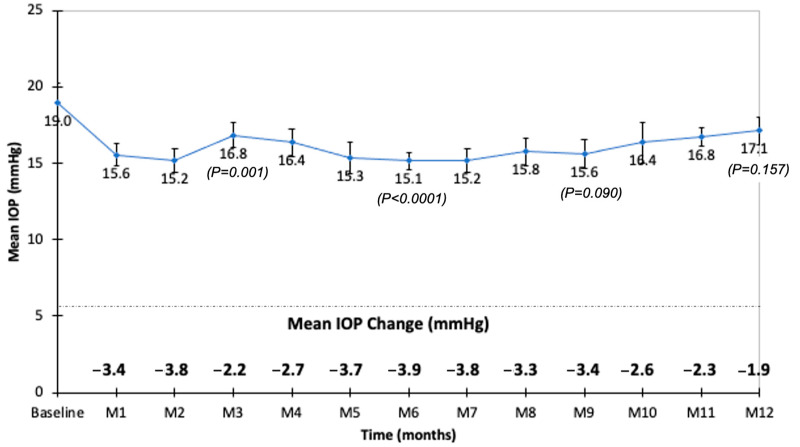
Mean intraocular pressure (IOP) and mean change in IOP during the follow-up after fluocinolone acetonide intravitreal injection.

**Figure 6 pharmaceutics-14-02391-f006:**
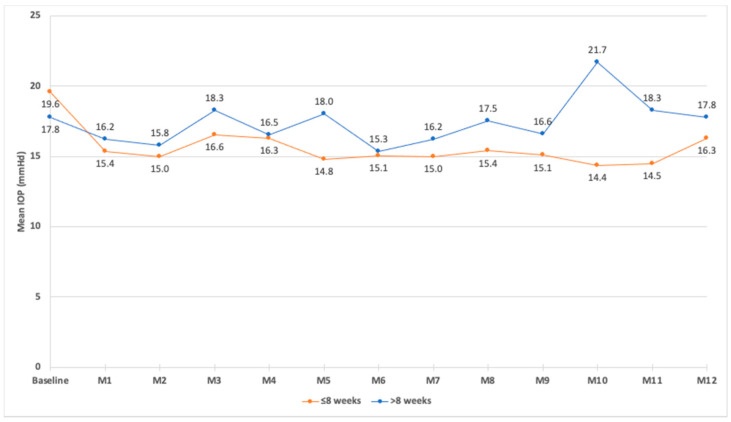
Mean intraocular pressure (IOP) during the follow-up in eyes that received their last dexamethasone intravitreal implant injection ≤8 weeks or >8 weeks before the first injection of fluocinolone acetonide implant.

**Table 1 pharmaceutics-14-02391-t001:** Baseline demographics and ocular characteristics. BCVA, best-corrected visual acuity; CMT, central macular thickness; ERM, epiretinal membrane; ETDRS, early treatment diabetic retinopathy study; IOP, intraocular pressure; NPDR, non-proliferative diabetic retinopathy; PDR, proliferative diabetic retinopathy; PRP, panretinal photocoagulation; SD, standard deviation.

Baseline Demographics		n
Age, years (mean ± SD)	69.8 ± 10.2	107
Gender, male/female (n (%))	55 (48.7)/58 (51.3)	113
Type of diabetes, n (%)		108
Type 2	100 (89.3)	
Type 1	8 (7.1)	
Insulin therapy, n (%)	67 (62.6)	107
Diabetes duration, years (mean ± SD)	21.9 ± 12.7	93
HbA1c level, mean (%)	7.4 ± 0.1	80
High blood pressure, n (%)	71 (70.3)	101
Ocular characteristics		n
Laterality, n (%)		113
OD	61 (54)	
OS	52 (46)	
Diabetic macular edema duration, months (mean ± SD [range])	71.8 ± 48.2 [15–360]	104
PDR n (%)	6 (5.6)	108
NPDR, n (%)	32 (29.6)	108
PRP, n (%)	70 (64.8)	108
Pseudophakic, n (%)	103 (91.2)	113
IOP-lowering medications, n (%)	23 (25.6)	90
Monotherapy	12 (52.2)	
Dual therapy	9 (39.1)	
Triple therapy	2 (8.7)	
Prior vitrectomy, n (%)	30 (27.3)	110
Prior ERM peeling, n (%)	19 (18.1)	110
ERM, n (%)	14 (13.3)	105
BCVA, ETDRS letters (mean ± SD)	54.1 ± 17.8	113
CMT, μm (mean ± SD)	454.7 ± 196.7	108
IOP, mmHg (mean ± SD)	19.0 ± 4.5	74

**Table 2 pharmaceutics-14-02391-t002:** Treatments for diabetic macular edema received prior to fluocinolone acetonide intravitreal injection. DEX, dexamethasone implant; FAc, fluocinolone acetonide implant; SD, standard deviation.

Prior Treatments		n
Intravitreal DEX, n (%)	113 (100)	113
Number of DEX injections per eye (mean ± SD [range])	6.3 ± 4.5 [1–21]	113
Time between two DEX injections, weeks (mean ± SD)	14.8 ± 4.5	69
Time between the last DEX injection and the first FAc injection, weeks (mean ± SD [range])	11.12 ± 22 [1–163]	102
Macular laser therapy, n (%)	27 (26.2)	103
Intravitreal triamcinolone, n (%)	13 (11.5)	113
Peribulbar triamcinolone, n (%)	5 (4.4)	113
Intravitreal ranibizumab, n (%)	72 (63.7)	113
Number of ranibizumab injections per eye (mean ± SD [range])	7.9 ± 5.8 [2–23]	72
Intravitreal aflibercept, n (%)	49 (43.4)	113
Number of aflibercept injections per eye (mean ± SD [range])	7.53 ± 8.3 [1–36]	47
Intravitreal bevacizumab, n (%)	10 (8.9)	113
Number of bevacizumab injections per eye (mean ± SD [range])	1.6 ± 1.8 [1–6]	8

**Table 3 pharmaceutics-14-02391-t003:** Proportion of eyes with a visual-acuity gain or loss in ETDRS letters during the follow-up after fluocinolone acetonide implant injection. ETDRS, early treatment diabetic retinopathy study; VA, visual acuity.

	Month 1	Month 3	Month 6	Month 12
Number of eyes	67	46	44	20
VA gain ≥ 5 ETDRS letters	36%	41%	45%	40%
VA gain ≥ 10 ETDRS letters	19%	24%	32%	35%
VA gain ≥ 15 ETDRS letters	9%	15%	20%	20%
VA stability (±4 ETDRS letters)	42%	39%	27%	25%
VA loss ≥ 5 ETDRS letters	21%	17%	25%	0%

**Table 4 pharmaceutics-14-02391-t004:** Variables associated with the risk of needing additional treatments. BCVA, best-corrected visual acuity; CMT, central macular thickness; DEX, dexamethasone implant; FAc, fluocinolone acetonide implant.

	*p*
Diabetes mellitus duration	0.009
Diabetic macular edema duration	0.973
Age	0.552
Associated high blood pressure	0.927
HbA1c level	0.544
Stage of diabetic retinopathy	0.068
Baseline BCVA (prior to FAc injection)	0.599
Baseline CMT (prior to FAc injection)	0.003
Time between the last DEX injection and the first FAc injection	0.035
Number of prior DEX injections	0.273

## Data Availability

The data presented in this study are available on request from the corresponding author. The data are not publicly available due to the French legislation regarding the protection of personal data (CNIL Commission Nationale de l’Informatique et des Libertés https://www.cnil.fr/en/personal-data-definition).

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
