# Peer review of "Management of Patients with Diabetic Macular Edema Switched from Dexamethasone Intravitreal Implant to Fluocinolone Acetonide Intravitreal Implant"

_pharmaceutics, 2022, doi:10.3390/pharmaceutics14112391_

Round 1

Reviewer 1 Report

In this multicenter study the authors analyzed the clinical efficacy and safety profile of fluocinolone acetonide (FAci)  after having switched from dexamethasone implant to in patients with DME. The manuscript is well written and structured, conclusions are supported by the results, and the topic is appealing for the readers. Moreover, the sample size is relatively large. There are some issues to be addressed:
-Firstly, an English revision by a native speaker should be performed.
- In the introduction, please cite the following article while dealing with the sub-optimal outcomes in real-life studies with anti-VEGF agents (PMID 
32252053)
- In the methods, the authors should specify the exclusion criteria. Did they rule out other possible causes of IRF, including a medical history of uveitis or other macular disorders? Please, clarify this point.
- In the methods, they include also patients with ERM. Do
the authors consider that this could be a possible confounding factor while measuring CMT and BCVA, in comparison with patients without a medical history of VMT?
- In the discussion, the authors should discuss the concept that they only considered CMT as anatomical outcome in patients treated with FAci; however, recent studies have highlighted that CMT is an 'obsolete' parameter while considering the retinal anatomy. In fact, other parameters such as the ONL layer thickness and the RPE+photoreceptor layer thickness, the variations in volumes of IRF and SRF are more accurate indicators of BCVA changes in patients with macular exudative disorders. Please discuss this important point.

Reviewer 2 Report

In this research work, the authors evaluated the efficacy and safety of FAc intravitreal implants in patients with chronic DME previously treated with DEX intravitreal implants. Although this work is within the scope of "Pharmaceutics", it needs major revision before publication in this journal. I have the following comments and/or suggestions:

-         -What is the novelty of this study?

-      -   Rewrite the title. It is too long and not concise enough.

-     -Check the authors list. You have a mixture of small letters and capital letters. Also, delete “and on behalf of the CFSR GROUP”, you have to be precise with the authorship. Moreover, delete the keyword “switch”.

-     - The abstract is not clear. Firstly, you have to start explaining the aim of this study, and then describe the methods. Regarding the methods for example, morphologic response to dexamethasone was defined based on central macular thickness (CMT) changes after the DEXi treatment. Steroid-response was defined as intraocular pressure (IOP) elevation ≥5 mmHg or IOP > 21 mmHg after any previous DEX exposure…. Then, you follow with the results.

-       - The references. Please, you have to add the DOI of each reference. Also, in the section “Instructions for Authors” you have the format and style that you have to follow for the references. Please follow that guidelines, in the case of journal articles is this:

1. Author 1, A.B.; Author 2, C.D. Title of the article. Abbreviated Journal Name Year, Volume, page range.
